# Exploring loneliness: Prevalence and determinants among medical students in Bangladesh

Zarin Tasnim Maliha[1¤a]*, Sayeda Nazmun Nahar[1¤b], Dipak Kumar Mitra[1],
Nadira Sultana Kakoly[1], Kamrun Nahar Koly[2], Md Humayun Kabir Talukder[3],
Helal Uddin Ahmed[4], Rajat Das Gupta[5], M. Tasdik Hasan[6]

1 Department of Public Health, North South University, Dhaka, Bangladesh, 2 Health System and Population Studies Division, International Centre for Diarrhoeal Disease Research, Bangladesh (icddr,b), Dhaka, Bangladesh, 3 Centre for Medical Education (CME), Dhaka, Bangladesh, 4 Department of Child Adolescent and Family Psychiatry, Faridpur Medical College, Faridpur, Bangladesh, 5 Department of Epidemiology and Biostatistics, Arnold School of Public Health, University of South Carolina, Columbia, South Carolina, United States of America, 6 Action Lab, Faculty of Information Technology, Monash University, Melbourne, Australia

¤a Current address: Technical Unit, RTM International, Dhaka, Bangladesh
¤b Current address: Department of Women's and Child Health, Uppsala University, Uppsala, Sweden
* zmaliha10@gmail.com

## Abstract

Loneliness, a deeply distressing emotional state, has emerged as a critical factor contributing to mental health challenges among medical students. It is associated with poor mental and physical health outcomes, and its prevalence among medical students is rising globally. Past studies indicated that doctors are among the loneliest professionals. This study aims to assess the prevalence of loneliness and its associated factors among medical students in Bangladesh. To the best of our knowledge, this is the first scientific investigation to evaluate loneliness among medical students in the country. A cross-sectional study was conducted in 2021 among 529 current medical students (1st to 5th year) across Bangladesh using an online survey and convenience sampling techniques. The UCLA Loneliness Scale was used to assess levels of loneliness, along with relevant socio-demographic questionnaires. Among the participants, 27.3% reported a high degree of loneliness, while 52.2% experienced moderate levels of loneliness. The remaining 20.5% reported low levels of loneliness. Unhealthy lifestyle factors such as smoking, drug abuse, and other risky habits were found to be prevalent among participants experiencing loneliness. The findings underscore the urgent need to establish robust mental health services in medical colleges to address the mental well-being of students. Given the limitations of this study, including its cross-sectional nature, and the prolonged impact of the COVID-19 pandemic on students' mental health, further longitudinal and in-depth studies are recommended to explore this issue comprehensively.

**Data availability statement:** The minimal dataset underlying the findings of this study has been uploaded as Supporting Information. The dataset includes anonymized numeric responses to the UCLA Loneliness Scale along with selected demographic variables required to replicate the analyses. No personally identifiable information is included.

**Funding:** This study was made possible through funding from the Centre for Non-Communicable Diseases and Nutrition, a collaborative center of excellence at the BRAC James P Grant School of Public Health. The grant was supported by the Global Health Research Unit (GHRU) based at Imperial College London (ICL). The funding support of the GHRU is from the National Institute of Health Research, UK (NIHR). The funders had no role in study design, data collection and analysis, decision to publish, or preparation of the manuscript. No authors received a salary from these funders.

**Competing interests:** I have read the journal's policy and the authors of this manuscript have the following competing interests: 1. We have received funding from the Centre for Non-Communicable Diseases and Nutrition (CNCDN) for this research. 2. We declare no other financial and non-financial interests related to this manuscript.

## Introduction

Medical schools play a critical role worldwide in equipping students with the knowledge and skills required to produce competent doctors. They ensure that the medical curriculum delivers a comprehensive blend of education and practical training [1]. However, the demanding nature of medical education often places immense pressure on students, leading to significant mental health challenges. Medical students are particularly vulnerable to stress, anxiety, and depression compared to their peers in non-medical disciplines [2,3]. The rigorous academic workload, frequent examinations, long training hours, and societal expectations often push students to the brink, leading to frustration, anxiety, depressive symptoms, and even suicidal ideation. Globally, studies have revealed alarming trends: in China, 29% of medical students experienced depression, 21% anxiety, and 11% suicidal ideation [4]; similarly, the British Medical Association reported six suicides among medical students in 2018 alone [5]. In Bangladesh, where the medical curriculum consists of five years of study followed by a year-long internship [6], the situation is equally concerning.

Bangladesh ranks among the nations with the highest proportion of medical students experiencing mental health disorders. A cross-sectional study in Dhaka found that 38.9% of medical students reported depression, and 17.6% had suicidal thoughts [3]. Another study revealed that 33.5% of medical students suffered from mental health issues—nearly double the prevalence seen in the general adult population (16.8%) as reported in the National Mental Health Survey 2018/19. Moreover, the prevalence of depression and suicidal tendencies among Bangladeshi medical students exceeds rates seen in other countries, such as Cameroon (30%) and the global average of 33.6% [2,7].

The mental health crisis among medical students can largely be attributed to the demanding curriculum, socio-economic pressures, and lack of a supportive learning environment [3,8]. Stress exacerbates existing vulnerabilities, particularly among younger students living away from home, adjusting to new environments, and grappling with societal expectations. In addition, ingrained cultural practices in Bangladesh discourage students from seeking professional help due to the stigma surrounding mental health care [9,10]. Fear of being perceived as weak or unfit for the profession prevents many students from accessing essential support, worsening their emotional distress and academic struggles.

Loneliness is the subjective feeling of isolation, lack of belonging, or absence of meaningful companionship. It is recognized as an emotional and mental state that may manifest as psychological distress, including sadness, low self-esteem, and hopelessness, particularly when individuals perceive a gap between their desired and actual social connections [11,12]. Loneliness, a deeply distressing emotional state, has emerged as a critical factor exacerbating mental health issues among medical students. Loneliness is associated with poor mental and physical health outcomes, and its prevalence among medical students is rising globally. A recent study found that doctors are among the loneliest professionals [13]. In Bangladesh, the lack of mental health infrastructure further compounds these challenges. While the U.S. has one psychiatrist for every 7,500 people, Bangladesh has an alarming shortage, with

only 0.4% of healthcare professionals working in mental health [14]. Consequently, access to mental health support for medical students remains limited.

This study aims to shed light on the prevalence of loneliness and its associated factors among medical students in Bangladesh. By identifying key contributors, the findings will provide actionable insights to bridge gaps in mental health support addressing loneliness among medical students and inform future interventions. This study is, to the best of our knowledge, the first scientific investigation to assess loneliness among medical students in Bangladesh.

## Methods

### Ethics statement

The study obtained ethical approval (IRB Reference no: 2020/OR-NSU/IRB-No.0502) from North South University, Bangladesh. Electronic informed consent was obtained from all participants prior to data collection. Privacy and confidentiality of participants were maintained throughout the study to ensure the ethical standards were strictly adhered to.

### Study design and setting

This study employed a cross-sectional design to assess loneliness, among medical students in Bangladesh. Data collection was conducted from 22nd January, 2021–12th February, 2021. The study targeted currently enrolled Bachelor of Medicine & Bachelor of Surgery (MBBS) and Bachelor of Dental Surgery (BDS) students from both public and private medical colleges across Bangladesh. The study population consisted of Bangladeshi medical students (1st to 5th year) from 28 public medical colleges and 29 private medical colleges. The inclusion criteria were as follows:

- Bangladeshi students,
- Not clinically diagnosed with any chronic disease,
- Not clinically diagnosed with chronic depression,
- Not known to have a history of drug abuse.

**Sampling method and sample size.** A convenience sampling method was employed to select participants. Initially, a sample size of 384 students was targeted. However, data were collected from 529 students after approaching around 600 students. The oversampling was necessary due to the transition from face-to-face to online data collection because of the COVID-19 pandemic.

**Instruments used.** Loneliness was measured using the validated UCLA Loneliness Scale, a widely used tool for assessing subjective feelings of loneliness. The data collection was conducted online through surveys, and all participants who met the inclusion criteria and agreed to participate were electronically consented before the interview.

**Data management and analysis.** Data analysis was performed using SPSS software (version 22.0). Descriptive statistics, including central tendency (mean, frequency, percentage) and dispersion (standard deviation, range), were calculated for participants' demographic characteristics. For the relationship between categorical variables, a Chi-square test was applied. For continuous variables, the two-sample independent t-test was used. A p-value of less than 0.05 was considered statistically significant for all tests.

**Quality control and assurance.** Periodic assessments of data quality, participant recruitment, and other influencing factors were carried out by the Principal Investigator (PI). No interventions were applied during the study, and there was no follow-up mechanism due to resource limitations.

## Results

A total of 529 responses were included in the analysis. The majority of participants were female (58.8%). The highest proportion of participants were in their 3rd year of medical school (38.3%), followed by 5th-year students (22.8%) and

4th-year students (15.7%). The mean age of the participants was 22.75 years (± 1.98), with most students falling between the ages of 18–24 years (81.6%). Regarding living arrangements, most participants resided in hostels (61.1%), while 34.2% lived with their family members. Among lifestyle habits, smoking was the most common, with 18% of participants identifying as smokers. Additionally, 11.2% consumed caffeine, 4.9% were alcohol users, and 3.4% reported using substances. In terms of parental education, 44.4% of fathers and 28.8% of mothers had completed postgraduate education. The majority of participants (56.2%) and their family members (40.8%) had 4 or fewer family members. Over 34% of participants reported having elderly family members. The economic status of the participants' families was relatively strong compared to the national average. Approximately 35% of participants had a monthly family income between 25,001–50,000 Bangladeshi Taka (BDT), 30% had incomes ranging from 50,001–99,999 BDT, and more than 15% reported a monthly family income of 100,000 BDT or higher. Around 20% of participants reported having personal income, 12.3% contributed financially to their families, and 67.7% received pocket money from various sources (Table 1).

**Prevalence of loneliness among the participants**

According to the study findings, 27.3% participants were facing high degrees of loneliness, 29.4% were facing moderately high degrees of loneliness, 22.8% were facing moderate degrees of loneliness, and the remaining 20.5% were facing low degrees of loneliness (Fig 1).

**Association between sociodemographic characteristics and degrees of loneliness**

Among the socio-demographic factors, degrees of loneliness had statistically significant association with age ($p = 0.000$), living place ($p = 0.04$), smoking habit ($p < 0.001$), substance abuse ($p = 0.001$), fathers' education ($p = 0.00$) and mother's education ($p = 0.00$) (Table 2).

## Discussion

Our study highlighted the levels of loneliness among medical students in Bangladesh and identified several contributing factors. Among the health-related habits examined, smoking emerged as the most prevalent, followed by caffeine consumption, alcohol use, and substance abuse. This aligns with findings from a recent study among medical students, which also reported smoking as the most common habit, followed by alcohol consumption. Promoting a healthy lifestyle among medical students is crucial, as they are future healthcare professionals who need to prioritize their own health and serve as role models for patients [15].

The educational level of participants' parents was notably higher than the national average in Bangladesh. This suggests that children of educated parents are more likely to pursue higher education and careers in medicine [16]. Additionally, with Bangladesh's aging population steadily increasing—now exceeding 13 million people aged 60 and older [17]—our study revealed that more than one-third of participants had elderly family members. The relatively better economic status of participants' families, compared to the national average, can be attributed to their parents' higher education levels and socio-economic backgrounds [18].

Previous studies have identified doctors as being among the loneliest professionals [13]. In our study, 56% of participants reported moderate to high levels of loneliness, a figure notably higher than findings from studies conducted in Ethiopia [19], Turkey [20], and Thailand [21]. Several socio-demographic factors were significantly associated with loneliness in our sample, including age, gender, year of education, smoking habits, caffeine consumption, mother's education, presence of elderly family members, and receipt of pocket money. These results are comparable to findings from a study in South India, where factors such as year of study, extracurricular participation, family history of depression, financial stress, substance abuse, romantic relationships, family problems, and health issues were strongly linked to depression [22]. Other studies have also demonstrated a significant correlation between loneliness and depression [23,24].

**Table 1. Sociodemographic characteristics of the study participants.**

| Variables with Categories | Frequency (No.) | Percent (%) |
|---|---|---|
| **Age Group** | | |
| 18–24 years | 430 | 81.6 |
| 25–29 years | 97 | 18.4 |
| **Gender** | | |
| Female | 310 | 58.8 |
| Male | 212 | 40.2 |
| Not disclosed | 5 | 0.9 |
| **Year of education** | | |
| 1st year medical student | 70 | 13.3 |
| 2nd year medical student | 52 | 9.9 |
| 3rd year medical student | 202 | 38.3 |
| 4th year medical student | 83 | 15.7 |
| 5th year medical student | 120 | 22.8 |
| **Living Place** | | |
| Hostel | 322 | 61.1 |
| Mess | 18 | 3.4 |
| With Family | 180 | 34.2 |
| Others | 7 | 1.3 |
| **Common habits** | | |
| Smoking | 95 | 18 |
| Alcoholism | 26 | 4.9 |
| Caffeine | 59 | 11.2 |
| Substance abuse | 18 | 3.4 |
| **Father's Education** | | |
| Primary | 11 | 2.1 |
| Secondary | 55 | 10.4 |
| Higher Secondary | 109 | 20.7 |
| Undergraduate | 94 | 17.8 |
| Post-graduate | 234 | 44.4 |
| Others | 24 | 4.6 |
| **Mother's Education** | | |
| Primary | 18 | 3.4 |
| Secondary | 99 | 18.8 |
| Higher Secondary | 134 | 25.5 |
| Undergraduate | 104 | 19.7 |
| Post-graduate | 152 | 28.8 |
| Others | 20 | 3.8 |
| **Number of members in family** | | |
| ≤4 members | 215 | 40.8 |
| 5–8 members | 296 | 56.2 |
| ≥9 members | 16 | 3.0 |
| **Presence of aged family members in family** | | |
| Yes | 182 | 34.5 |
| No | 345 | 65.5 |

*(Continued)*

**Table 1.** (Continued)

| Variables with Categories | Frequency (No.) | Percent (%) |
|---|---|---|
| **Monthly total family income (BDT)** | | |
| ≤25000 | 109 | 20.7 |
| 25001-50000 | 183 | 34.7 |
| 50001-99999 | 155 | 29.4 |
| ≥100000 | 80 | 15.2 |
| **Personal income** | | |
| Yes | 105 | 19.9 |
| No | 422 | 80.1 |
| **Family contribution** | | |
| Yes | 65 | 12.3 |
| No | 462 | 87.7 |
| **Get pocket money from various source** | | |
| Yes | 357 | 67.7 |
| No | 170 | 32.3 |

Loneliness is a significant concern across different demographic groups, with varying prevalence rates reported depending on age, setting, and context. Among graduate university students, a study conducted in Bangladesh reported that approximately 43% experienced loneliness, indicating a substantial mental health burden even among young and educated populations [25]. In contrast, older adults tend to report even higher levels of loneliness. One international study that included individuals aged 60 years and above found that more than half (51.5%) of participants experienced loneliness, highlighting the emotional vulnerability of aging populations [26]. Similarly, a longitudinal survey analyzing loneliness

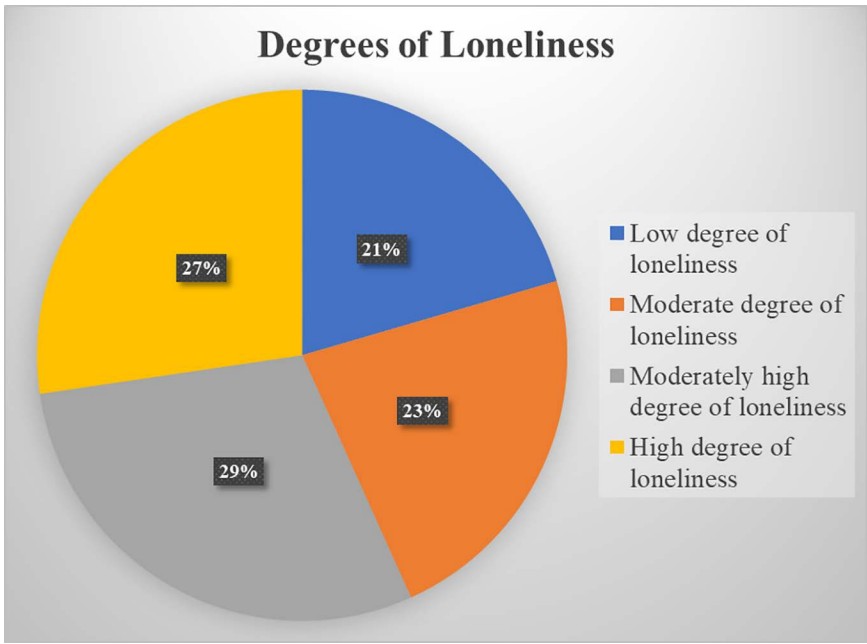

**Fig 1. Degrees of loneliness according to UCLA loneliness scale.** Distribution of loneliness degrees among medical students: high, moderately high, moderate and low.

**Table 2. Association between sociodemographic characteristics and degrees of loneliness of study participants.**

| Variables with Categories | Degrees of loneliness | | | | Chi-square | P value |
|---|---|---|---|---|---|---|
| | Low (%) | Moderate (%) | M-high (%) | High (%) | | |
| **Age** | | | | | | |
| 18–24 years | 101(23.5) | 110(25.6) | 113(26.3) | 106(24.7) | 32.23 | 0.000* |
| 25–39 years | 7(7.2) | 10(10.3) | 42(43.3) | 38(39.2) | | |
| **Gender** | | | | | | |
| Female | 65(21.0) | 79 (25.5) | 86(27.7) | 80(25.5) | 10.61 | 0.101 |
| Male | 43(20.3) | 38(17.9) | 69(32.5) | 62(29.2) | | |
| Not disclosed | 0(0.00 | 3(60.0) | 0(0.0) | 2(40.0) | | |
| **Year of education** | | | | | | |
| 1st year | 17(24.3) | 17(24.3) | 20(28.6) | 16(22.9) | 15.48 | 0.216 |
| 2nd year | 8(15.4) | 9(17.3) | 20(38.5) | 15(28.8) | | |
| 3rd year | 47(23.3) | 56(27.7) | 53(26.2) | 46(22.8) | | |
| 4th year | 18(21.7) | 14(16.9) | 23(27.7) | 28(33.7) | | |
| 5th year | 18(15.0) | 24(20.0) | 39(32.5) | 39(32.5) | | |
| **Living Place** | | | | | | |
| Hostel | 63(19.6) | 72(22.4) | 111(34.5) | 76(23.6) | 17.6 | 0.04* |
| Mess | 2(11.1) | 2(11.1) | 7(38.9) | 7(38.9) | | |
| With Family | 42(23.3) | 45(25.0) | 35(19.4) | 58(32.2) | | |
| Others | 1(14.3) | 1(14.3) | 2(28.6) | 3(42.9) | | |
| **Smoking** | | | | | | |
| No | 100(23.1) | 111(25.7) | 115(26.6) | 106(24.5) | 30.4 | 0.00* |
| Yes | 8(8.4) | 9(9.5) | 40(42.1) | 38(40.0) | | |
| **Alcoholism** | | | | | | |
| No | 106(21.2) | 117(23.4) | 143(28.5) | 135(26.9) | 6.84 | 0.077 |
| Yes | 2(7.7) | 3(11.5) | 12(46.2) | 9(34.6) | | |
| **Caffeine use** | | | | | | |
| No | 100(21.4) | 108(23.1) | 137(29.3) | 123(26.3) | 3.42 | 0.33 |
| Yes | 8(13.6) | 12(20.3) | 18(30.5) | 21(35.6) | | |
| **Substance abuse** | | | | | | |
| No | 107(21.0) | 119(23.4) | 142(27.9) | 141(27.7) | 16.84 | 0.001* |
| Yes | 1(5.6) | 1(5.6) | 13(72.2) | 3(16.7) | | |
| **Father's Education** | | | | | | |
| Primary | 3(27.3) | 0(0.0) | 5(45.5) | 3(27.3) | 51.31 | 0.00* |
| Secondary | 8(14.5) | 8(14.5) | 21(38.2) | 18(32.7) | | |
| Higher Secondary | 9(8.3) | 15(13.8) | 49(45.0) | 36(33.0) | | |
| Undergraduate | 19(20.2) | 26(27.7) | 27(28.7) | 22(23.4) | | |
| Post-graduate | 66(28.2) | 62(26.5) | 49(20.9) | 57(24.4) | | |
| Others | 3(12.5) | 9(37.5) | 4(16.7) | 8(33.3) | | |
| **Mother's Education** | | | | | | |
| Primary | 1(5.6) | 2(11.1) | 8(44.4) | 7(38.9) | 48.8 | 0.00* |
| Secondary | 18(18.2) | 24(24.2) | 35(35.4) | 22(22.2) | | |
| Higher Secondary | 15(11.2) | 19(14.2) | 48(35.8) | 52(38.8) | | |
| Undergraduate | 31(29.8) | 26(25.0) | 22(21.2) | 25(24.0) | | |
| Post-graduate | 38(35.0) | 47(30.9) | 38(25.0) | 29(19.1) | | |
| Others | 5(25.0) | 2(10.0) | 4(20.0) | 9(45.0) | | |

*(Continued)*

**Table 2.** (Continued)

| Variables with Categories | Degrees of loneliness | | | | Chi-square | P value |
|---|---|---|---|---|---|---|
| | Low (%) | Moderate (%) | M-high (%) | High (%) | | |
| **Number of members in family** | | | | | | |
| ≤4 members | 55(25.6) | 54(25.1) | 49(22.8) | 57(26.5) | 12.33 | 0.055 |
| 5–8 members | 51(17.2) | 64(21.6) | 100(33.8) | 81(27.4) | | |
| ≥9 members | 2(12.5) | 2(12.5) | 6(37.5) | 6(37.5) | | |
| **Presence of aged family members in family** | | | | | | |
| Yes | 38(20.9) | 38(20.9) | 54(29.7) | 52(28.6) | 0.62 | 0.89 |
| No | 70(20.3) | 82(23.8) | 101(29.3) | 92(26.7) | | |
| **Monthly total family income (BDT)** | | | | | | |
| ≤25000 | 21(19.3) | 18(16.5) | 40(36.7) | 30(27.5) | 16.41 | 0.059 |
| 25001-50000 | 26(14.2) | 44(24.0) | 54(29.5) | 59(32.2) | | |
| 50001-99999 | 43(27.7) | 36(23.2) | 40(25.8) | 36(23.2) | | |
| ≥100000 | 18(22.5) | 22(27.5) | 21(26.3) | 19(23.8) | | |
| **Personal income** | | | | | | |
| Yes | 19(18.1) | 23(21.9) | 38(36.2) | 25(23.8) | 3.05 | 0.384 |
| No | 89(21.1) | 97(23.0) | 117(27.7) | 119(28.2) | | |
| **Family contribution** | | | | | | |
| Yes | 12(18.5) | 13(20.0) | 22(33.8) | 18(27.7) | 0.899 | 0.826 |
| No | 96(20.8) | 107(23.2) | 133(28.8) | 126(27.3) | | |
| **Get pocket money from various source** | | | | | | |
| Yes | 73(20.4) | 84(23.5) | 107(30.0) | 93(26.1) | 1.057 | 0.788 |
| No | 35(20.6) | 36(21.2) | 48(28.2) | 51(30.0) | | |

trends during the COVID-19 pandemic revealed a high baseline prevalence of 51.5% in 2020, which declined modestly to 45.7% in 2021, though nearly half of the respondents continued to report loneliness, suggesting persistent challenges despite temporal improvements [27].

Focusing on the context of Bangladesh, a community-based study among older adults found the prevalence of loneliness to be 54.3%, indicating that loneliness among elderly individuals in Bangladeshi societies may be particularly pronounced due to factors such as limited social support, declining health, and intergenerational separation [28]. These findings collectively demonstrate that loneliness is not restricted to one particular age group or setting, but rather presents a widespread public health issue. The consistently high prevalence across studies underscores the need for context-specific interventions aimed at reducing social isolation and enhancing emotional well-being across the life course.

Loneliness is a profoundly distressing emotional experience with negative impacts on both physical and mental health, particularly when coupled with an unhealthy lifestyle [29]. Our analysis further revealed significant associations between loneliness and various socio-demographic factors, including age, living situation, smoking, substance abuse, and the education levels of both parents. These findings are consistent with previous research, which identified factors such as age, living alone, social interactions, psychological distress, social exclusion, education level, and employment status as being significantly associated with loneliness [30].

## Conclusion

This study identifies a high prevalence of loneliness among medical students in Bangladesh and underscores its association with a range of socio-demographic and lifestyle factors. The findings contribute to the growing body of literature on mental health concerns in medical education and offer critical insights into the psychological vulnerabilities of this

population. Nevertheless, certain methodological limitations should be acknowledged, including the reliance on online data collection and the use of convenience sampling, which may affect the generalizability of the results. Despite these constraints, the study underscores the pressing need for comprehensive mental health support mechanisms within medical institutions in Bangladesh. Evidence-based interventions- such as accessible counseling services, structured stress management programs, and the incorporation of mental health literacy into the medical curriculum- should be prioritized to enhance the psychological well-being and academic performance of medical students, ultimately contributing to a more resilient future healthcare workforce.

## Limitations of the study

The study faced several limitations that should be acknowledged. First, due to pandemic-related risks, face-to-face interviews were not feasible, which may have influenced the depth and quality of responses. Consequently, the findings should be interpreted with caution, particularly given the convenience sampling techniques, limited sample size and potential lack of broader representation. Additionally, the study only included participants with internet access, which may have excluded individuals from underprivileged or rural backgrounds, thereby limiting the diversity of the sample. The study did not include multivariate analysis to control for confounding variables, nor did it calculate effect sizes or test for normality assumptions prior to conducting t-tests. This limits the ability to infer independent associations and may affect the robustness and generalizability of the findings. Future studies with more rigorous statistical methods are recommended to validate these associations.

## Recommendations

Addressing the significant prevalence of loneliness among medical students in Bangladesh requires urgent and coordinated action from academic institutions, healthcare policymakers, and stakeholders. The findings of this study highlight the critical need for a supportive and nurturing environment for these future healthcare providers. One key initiative is the establishment of mental health support systems within medical institutions. Counseling centers staffed with trained mental health professionals should be created to provide confidential and accessible support to students. Additionally, peer-led mental health programs and structured workshops focusing on stress management, emotional resilience, and coping strategies could offer both immediate and preventive assistance, equipping students with the tools they need to navigate their academic and personal challenges. Leveraging technology can further extend the reach and accessibility of mental health support. Online counseling platforms, mental health apps, and telehealth services can provide valuable resources, particularly for students in remote or underserved areas. Regular monitoring and the implementation of tailored interventions that address gender-specific needs are essential to effectively support the evolving mental health challenges faced by medical students. Institutions should conduct anonymous surveys to assess these needs and identify specific challenges. The insights gained from such assessments can guide the development of targeted interventions designed to address the unique pressures faced by medical students. By implementing these strategies, academic institutions can foster an inclusive and holistic support system that prioritizes mental health, strengthens resilience, and equips students to thrive as compassionate and effective healthcare professionals.

## Supporting information

**S1 Data.  Dataset.**
(XLSX)

## Acknowledgments

We would like to extend our heartfelt gratitude to the volunteers who assisted with data collection, ensuring the smooth progression of this study. Their dedication and commitment were instrumental in overcoming challenges, particularly during the unprecedented circumstances of the pandemic.

## Author contributions

**Conceptualization:** Zarin Tasnim Maliha, Sayeda Nazmun Nahar, M. Tasdik Hasan.

**Data curation:** Zarin Tasnim Maliha.

**Formal analysis:** Zarin Tasnim Maliha.

**Funding acquisition:** Sayeda Nazmun Nahar.

**Investigation:** Zarin Tasnim Maliha, Sayeda Nazmun Nahar.

**Methodology:** Zarin Tasnim Maliha, M. Tasdik Hasan.

**Project administration:** Zarin Tasnim Maliha, Sayeda Nazmun Nahar.

**Resources:** Helal Uddin Ahmed, M. Tasdik Hasan.

**Software:** Zarin Tasnim Maliha.

**Supervision:** Dipak Kumar Mitra, Nadira Sultana Kakoly, Kamrun Nahar Koly, Md Humayun Kabir Talukder, Helal Uddin Ahmed, Rajat Das Gupta, M. Tasdik Hasan.

**Visualization:** Zarin Tasnim Maliha.

**Writing – original draft:** Zarin Tasnim Maliha.

**Writing – review & editing:** Sayeda Nazmun Nahar, Dipak Kumar Mitra, Nadira Sultana Kakoly, Kamrun Nahar Koly, Md Humayun Kabir Talukder, Helal Uddin Ahmed, Rajat Das Gupta, M. Tasdik Hasan.

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
