## [Decision Letter · Decision Letter 0]

3 Mar 2025

PMEN-D-25-00014

Exploring loneliness: Prevalence and determinants among medical students in Bangladesh

PLOS Mental Health

Dear Dr. Zarin Tasnim Maliha

Thank you for submitting your manuscript to PLOS Mental Health. After careful consideration, we feel that it has merit but does not fully meet PLOS Mental Health’s publication criteria as it currently stands. Therefore, we invite you to submit a revised version of the manuscript that addresses the points raised during the review process.

EDITOR: I would request authors to consider the comments by the peer reviewers, especially those around:

1. Data sharing. I agree with the reviewers that anonymized data should be possible to share. The authors should consider doing this and check with their ethics committee/institutional review board how they can go about it.

2. Statistical issues. Once again, I agree with the reviewers that the work requires more careful statistical analyses and grounded language around inference.

We look forward to receiving your revised manuscript.

Kind regards,

Siddhesh Zadey

Academic Editor

PLOS Mental Health

Additional Editor Comments (if provided):

Reviewers' comments:

Reviewer's Responses to Questions

**Comments to the Author**

1. Does this manuscript meet PLOS Mental Health’s publication criteria ? Is the manuscript technically sound, and do the data support the conclusions? The manuscript must describe methodologically and ethically rigorous research with conclusions that are appropriately drawn based on the data presented.

Reviewer #1: Partly

Reviewer #2: Yes

Reviewer #3: Yes

2. Has the statistical analysis been performed appropriately and rigorously?

Reviewer #1: No

Reviewer #2: No

Reviewer #3: Yes

3. Have the authors made all data underlying the findings in their manuscript fully available (please refer to the Data Availability Statement at the start of the manuscript PDF file)?

Reviewer #1: Yes

Reviewer #2: No

Reviewer #3: Yes

4. Is the manuscript presented in an intelligible fashion and written in standard English?

Reviewer #1: Yes

Reviewer #2: Yes

Reviewer #3: Yes

5. Review Comments to the Author

Reviewer #1: I enjoyed reading the manuscript and you make some very interesting observations.

There are several underlying issues that if addressed could bring additional clarity to the article:

1. There needs to be a clear definition of loneliness. This is critical to understand the rest of the paper. Without a working definition and parameters of this at the beginning, the understanding of the rest of the paper is limited.

2. There needs to be additional explanations regarding the statistical analysis used when considering the sociodemographic characteristics. You report significant findings, but without enough clarity to fully understand.

3. On line 173 your title is prevalence of depression, but this is inappropriate because it is not what you are measuring.

4. There needs to be more comparisons to the general population of Bangladesh. You make comparisons with some sociodemographic info, but it would be nice to see comparisons in loneliness (if they exist)

Reviewer #2: This study addresses an important and timely topic in mental health research. It highlights the high prevalence of loneliness among medical students and the association with various sociodemographic and behavioral risk factors.

Moreover, this study is well-structured and provides valuable insights. However, several methodological, statistical, and language-related concerns need to be addressed before the manuscript can be considered for publication in PLOS Mental Health. I encourage the authors to carefully revise their manuscript to enhance its impact and credibility. The statistical analysis is not fully rigorous due to the lack of multivariate analysis to control for Confounders variable, there is missing the effect sizes and unverified normality of the assumptions. Author need to ensure the effect sizes, the OR and CI reported. Also, check the normality assumptions before perform t-test. There is a lack of data availability statement due to ethical concerns but this need to be requested. The Conclusion and implication of this study need to imply the causal relationships rather than associations, lack of discussion on gender differences in loneliness, author need to suggest specific interventions to address loneliness among medical students.

Reviewer #3: The study demonstrates a technically sound approach with appropriate methodologies except using non probability sampling which affect generalizeablity. The conclusion reports that 56% of participants experienced moderate to high levels of loneliness and the various socio-demographic and lifestyle factors associated with loneliness which is supported by the findings of the study

6. PLOS authors have the option to publish the peer review history of their article (what does this mean? ). If published, this will include your full peer review and any attached files.

**Do you want your identity to be public for this peer review?** For information about this choice, including consent withdrawal, please see our Privacy Policy .

Reviewer #1: **Yes: ** Noah Hansen

Reviewer #2: **Yes: ** Samphoas Chien

Reviewer #3: **Yes: ** Ziyad Towfik Abdella

---

## [Editor Report · Decision Letter 1]

2 Jul 2025

Exploring loneliness: Prevalence and determinants among medical students in Bangladesh

PMEN-D-25-00014R1

Dear Dr Zarin Tasnim Maliha,

We are pleased to inform you that your manuscript 'Exploring loneliness: Prevalence and determinants among medical students in Bangladesh' has been provisionally accepted for publication in PLOS Mental Health.

Best regards,

Siddhesh Zadey

Academic Editor

PLOS Mental Health